# A Hybrid GA/ML-Based End-to-End Automated Methodology for Design Acceleration of Wireless Communications CMOS LNAs

Christos Sad *, Anastasios Michailidis *, Thomas Noulis and Kostas Siozios

Electronics Laboratory, Physics Department, Aristotle University of Thessaloniki, 54124 Thessaloniki, Greece; tnoul@physics.auth.gr (T.N.); ksiop@auth.gr (K.S.)
* Correspondence: csant@auth.gr (C.S.); anamicha@physics.auth.gr (A.M.)

**Abstract:** A new methodology for the RF/mmWave analog design process, automation and acceleration, is presented in this work. The proposed framework was implemented so as to accelerate the design cycle of analog/RF circuits by creating a dataset in a fully automated manner and training a combination of machine learning models for the optimal design parameters' prediction. machine learning polynomial regression was adopted to accelerate the design process, predicting the optimal design parameters' values while genetic algorithm optimization was exploited for the dataset creation automation. To evaluate the efficiency of the proposed methodology, the framework was implemented for the design of a common source Low-Noise-Amplifier, using a 65 nm CMOS process node. The proposed methodology successfully tackles the design cycle speed-up, automation, and acceleration, utilizing machine learning prediction for the design parameters and genetic algorithm for the dataset creation automation instead of the classical, simulation-based, standard design methodology. The provided experimental results have shown the effectiveness of the proposed hybrid approach, creating very precise RF matching networks for LNA designs and achieving >99.9% wave transmission efficiency while reaching >99% accuracy on the parameters' prediction task.

**Keywords:** meta-heuristics; machine learning; LNA design; analog design; automation; acceleration

## 1. Introduction

Complex analog circuit blocks are widely used in high speed communications SoC. This analog circuit complexity results in time-consuming issues across the analog design flow that even highly-experienced designers confront [1]. Furthermore, analog circuitry is designed according to very specific and, in some cases, strict specification criteria with respect to the targeted application [2]. Due to the rise of high-performance analog/mixed signal circuits and very narrow design error windows, analog design automation (ADA) and design acceleration have gained significant attention in various aspects of the analog design flow [3]. A very large gap separates the analog and digital design flows. In digital design flow, synthesis tools were efficiently developed to overcome the design complexity in a time-conservative manner but, in the analog design flow, only via simulations, the designers can confidently finalizing their schematics [4]. This lack of automated design frameworks has a time-consuming impact which is attached to every design step between schematic implementation and simulation until the full chip fabrication. To this end, the analog design flow strongly depends on the human factor, in a time-consuming non-automated manner.

A great number of recent studies have been focused on analog design automation and design cycle speedup. Some of them use modeling and simulation techniques and approaches that can be performed only inside the standard simulation suite [5,6]. In the majority, the design automation and acceleration methodologies rely on heuristics, meta-heuristics, and machine learning techniques optimization with the purpose of generating

new, alternative yet accurate, efficient, and fast design tools and methodologies outside the industry-standard simulation suite.

For example, in [7], an advanced genetic algorithm-based technique for the synthesis of microwave linear and low noise amplifiers is presented, allowing the automatic generation of amplifier schematics and passive element values directly from a set of performance and structural requirements while simultaneously selecting the best AE types/biases. In [8], taking advantage of the NSGA-II genetic algorithm, a design tool to determine the device sizes in an RF-LNA circuit was proposed, using HSPICE RF simulations to evaluate the fitness of the circuit specifications per every iteration of the GA. In [9], an application of an optimization technique using GA was proposed for LNAs, to optimize bias conditions at the simulation and design stage. It is mentioned that the technique can be extended to optimizing bias conditions in any active circuit, e.g., power amplifiers. Another automated design methodology was proposed in [10], where a genetic algorithm (GA), simulated annealing (SA) and the Levenberg–Marquardt (LM), working with a simulation tool, were tested to automate the design of a 18 GHz Low Noise Amplifier (LNA) using a 0.13 μm process node. The optimization tasks in this case were topology/matching dependant metrics, such as S-parameters, noise figure, and input-referred third-order intercept point. All the above examples take advantage of optimization techniques, in most cases the genetic algorithm and, by using a simulation tool for evaluation purposes, an automated simulation-based design tool is implemented. This process succeeds automation in the design flow but the searching operation (genetic algorithm's run) must be implemented for every desired new design. So, the designing process, in those particular cases, is not excessively accelerated.

On the other hand, machine learning (ML) approaches have also been widely used for analog design automation and design acceleration. In most cases, ML was introduced for the prediction of efficiency metrics for a design while, in some other cases, for design parameters prediction. For example, in [11], a feed-forward artificial neural network (ANN) for the synthesis of a radio-frequency, low noise amplifier (RF-LNA) circuit was proposed in combination with a GA for the optimal ANN architecture selection and input parameters tuning. Additionally, in [12], taking advantage of Bayesian linear regression and support vector machine models, an accelerated algorithm to explore the analog circuit performance limitations on the required technology was introduced. Finally, in [13], a hybrid machine learning and genetic algorithm automated design methodology was proposed for the automated synthesis of operational amplifiers. In this case, a neural network (NN) was trained, using data from simulation results, to predict efficiency metrics based on the circuit parameters, and then a GA in combination with the NN was employed to create optimal designs that meet the desired specifications. The GA uses the efficiency metrics, predicted from the NN, as a cost function. Although the ML approaches accelerate the design flow, the need for training data is obvious and, in most cases, simulation tools are employed for the dataset creation. Using design tools for the dataset creation is a human-based operation (non automated) with a time-consuming preprocess cost.

In conclusion, in the case of GA approaches for the design cycle automation, a design tool is employed to work synergistically with the GA and the provided results are concerning a single design, leading to the need for re-execution for new design specifications. This is in contrast with the use of ML where the results are generalized in numerous different design specifications and the predictions are extremely fast. However, the dataset creation methodology, in this particular scenario, is simulation-based (hand-crafted), resulting in lack of automation in a time-consuming way.

To tackle the above issues, a hybrid ML-GA automated design methodology was proposed. First, a genetic algorithm was employed to create, in a fully-automated manner, a dataset with sets of design descriptions. These design descriptions were to be used as input features in ML models. The optimal design parameters' values were used as targets for the ML models. Furthermore, a polynomial-regression ML model was introduced to be trained on the above dataset and predict the optimal design parameters' values

for any desired design from a pre-defined Design Space. The above methodology was implemented and tested for the design of a common source Low-Noise-Amplifier (LNA). The specific topology was selected due to its linear behavior (higher prediction accuracy) and low complexity with respect to application-dependant parameters (input matching network and transistor sizing). The low topology complexity was achieved by having only the operation frequency of the LNA as the crucial design specification, stepping aside the design requirements for the optimization of gain, noise figure (NF [14,15]), linearity (1 dB compression point [16,17]), and third order intercept point (IP3 point [18,19]).

## 2. LNA Topology and Impedance Matching Methodology

Low-noise-amplifiers or LNAs are tuned, high-frequency circuits used in wireless communication applications. When designing a low-noise-amplifier for high frequency applications, there are several concerns that need to be considered across the design flow. LNAs can be found in receiver blocks of mobile communication applications SoC [20,21] and their purpose is the amplification of the received signal without adding significant noise and compromising the overall signal-to-noise ratio [22]. This attribute of a low-noise-amplifier is derived from the input matching of the LNA at the desired frequency of interest which is a crucial specification with respect to the targeted application field. Furthermore, the frequency that the LNA matches is the driving factor of the design flow and leads to specific design parameters of the LNA topology using the same design process node, e.g., the sizing of the input transistor and the integrated spiral inductor values. Other design parameters, such as gain, noise figure, and linearity (1 dB compression point) depend on the designed topology (differential LNA, stacked FET LNA, multiple stages LNA, etc.), the inductor's quality factor, and the sizing of input and cascaded transistors.

One of the most fundamental theories that RF designers need to familiar with is the Network Theory and its relation to scattering parameters (S-parameters). The S-parameter matrix is a $N \times N$ matrix, where $N$ is the number of the network ports under test. If a network has two ports at its input and output as illustrated in Figure 1, a $2 \times 2$ S-parameter matrix can be derived which translates the relationship between the incident wave and the reflected and transmitted waves at each port of the network under test. The most essential scattering parameters for an RF network are the $S_{11}$ and the $S_{21}$ parameters which represent the reflection coefficient and the gain of the network, respectively. The $S_{11}$ parameters' representation of a two-port network describes the impedance mismatches between the source and load impedances and, according to the maximum power transfer theorem [23,24], as the impedance mismatch decreases, the incident wave is transmitted through the network and dissipated into the load with minimum wave reflections. Scattering parameters are frequency-dependent parameters and they use, in most cases, the decibels (dB) notation.

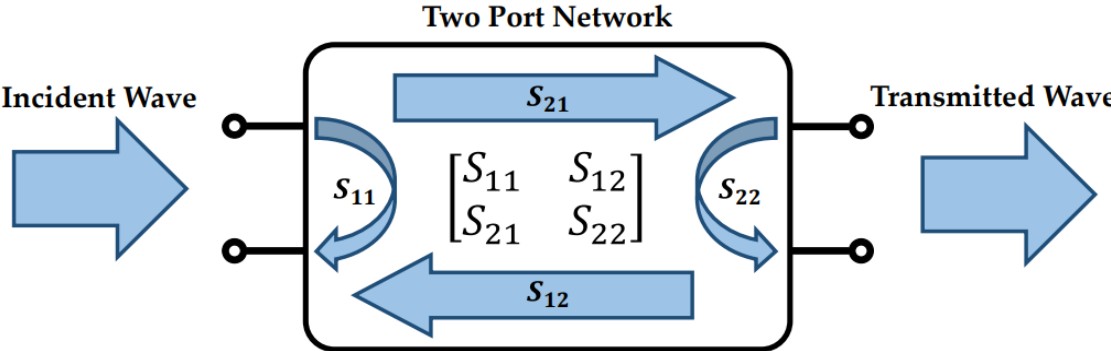

**Figure 1.** S-parameter matrix for a two-port RF network.

### 2.1. Voltage Standing Wave Ratio (VSWR)

The Voltage Standing Wave Ratio or VSWR is a measurement of the efficiency of the transmission power in RF/mmWave networks. Thus, the VSWR describes how efficiently the source power is transmitted and dissipated into the output load of the network [25]. The ratio between the transmitted and reflected waves in a network is often referred to as the Standing Wave Ratio (SWR). A high SWR indicates poor wave transmission, high wave reflections, and unmatched source and load impedances.

In ideal systems where the source and load impedances are exactly matched, the power transmission efficiency reaches 100%, which means that all of the source power can be delivered and dissipated into the output load with no wave reflections. Unfortunately, in real systems, the source and load impedances cannot be exactly matched to each other and, thus, the power transmission efficiency cannot reach 100%. Due to load mismatches, there are wave reflections compromising the power transmission efficiency. When the source and load impedances do not match, the incident and the reflected waves inside the network are superposed resulting in a standing wave. This voltage standing wave has its maximum and minimum values when the reflected wave is in-phase or 180° out-of-phase with the incident wave, respectively. These maximum and minimum values of the voltage standing wave can be used for the calculation of the VSWR [26],

$$VSWR = \frac{V_{max}}{V_{min}} \quad , \quad 1 \leq VSWR < \infty. \tag{1}$$

For ideal transmission, $V_{max} = V_{min}$ and the $VSWR = 1$. For real systems, almost 100% power transmission with negligible reflections can occur when $1 \leq VSWR \leq 1.05$. The VSWR measurement can be performed using the reflection coefficient $\Gamma$, which can be derived from the $S_{11}$ parameter as [27]

$$\Gamma = \frac{V_1^-}{V_1^+} = \frac{Z_L - Z_S}{Z_L + Z_S} \tag{2}$$

$$VSWR = \frac{1 + |\Gamma|}{1 - |\Gamma|}. \tag{3}$$

The VSWR measurements for low and high standing wave ratios are graphically illustrated in Figure 2. The high VSWR measurement depicted in Figure 2a represents source and load impedance mismatch while the low VSWR measurement represents the source and load impedance match of an RF network. Thus, in high VSWR representation, the reflected wave is comparable to the incident wave which indicates output load mismatch with respect to the source impedance. Due to the reflected wave, a standing wave is generated by the incident wave superposed with the reflected wave. The standing wave maximum value occurred when the incident and reflected wave are in-phase whereas its minimum value occurred when the incident and the reflected wave are 180° out-of-phase. By measuring the maximum and minimum values of the voltage standing wave, the VSWR of the system can be derived. In the low VSWR representation illustrated in Figure 2b, the reflection wave is significantly smaller than the incident wave, which indicates a better source-load impedance matching. The $V_{max}$ and $V_{min}$ values, in this case, are extremely close to each other resulting in a low $VSWR \approx 1$. When the VSWR is close to one, almost 100% of the source power can be delivered and dissipated into the output load.

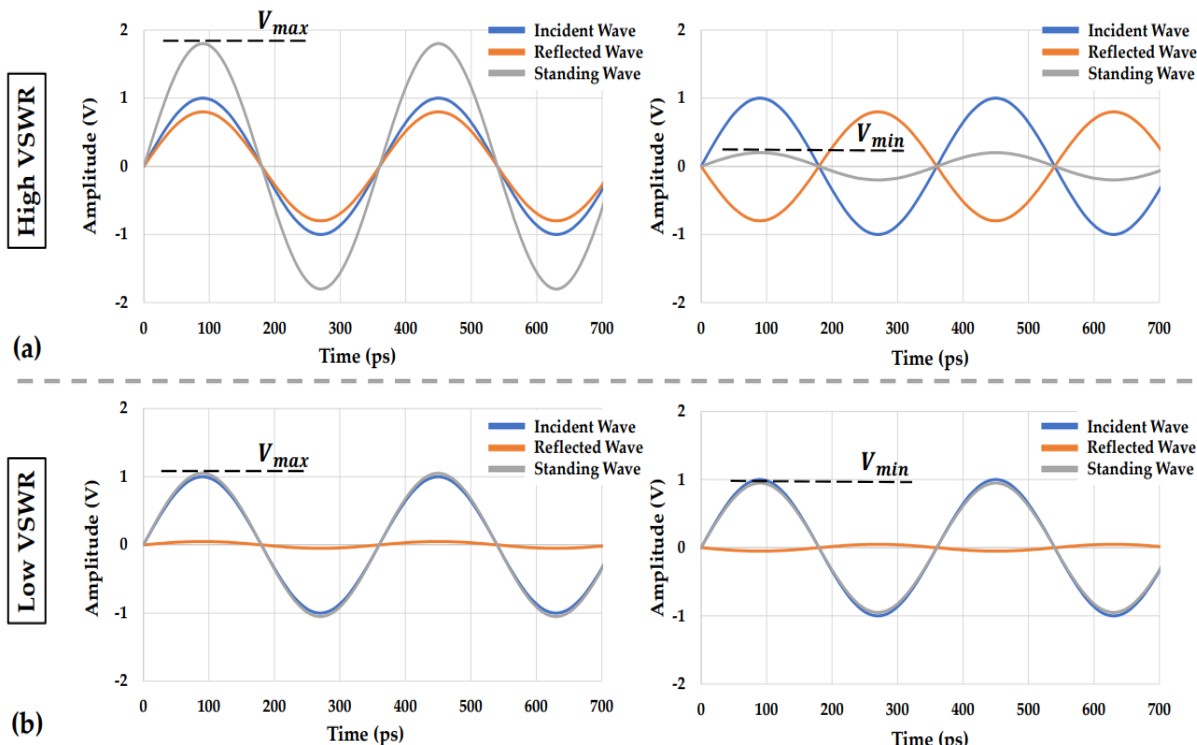

**Figure 2.** VSWR representation for (**a**) unmatched and (**b**) matched source and load impedances.

### 2.2. LNA Topology

The common source LNA topology is one of the most used topologies in RF/mmWave systems. Other topologies, such as differential LNAs or multi-stage LNA topologies, can be used when a low noise figure and high output gain need to be achieved. The common source LNA topology that was designed using a 65 nm RFCMOS process node is illustrated in Figure 3. The topology consists of two cascaded MOSFETs, M1 and M2, where M1 is the input transistor of the topology and the cascaded transistor M2 is used for gain boost and isolation between the input and output nodes of the topology. The input and output capacitors $C_1$ and $C_2$ were used as DC blocking capacitors. The $V_g$ voltage was applied, through a high value resistor, to the gate of the input transistor, setting it into the saturation region. The inductor $L_d$ was used as the output load of the topology whereas the inductors $L_g$ and $L_s$ were used for the input impedance matching of the LNA.

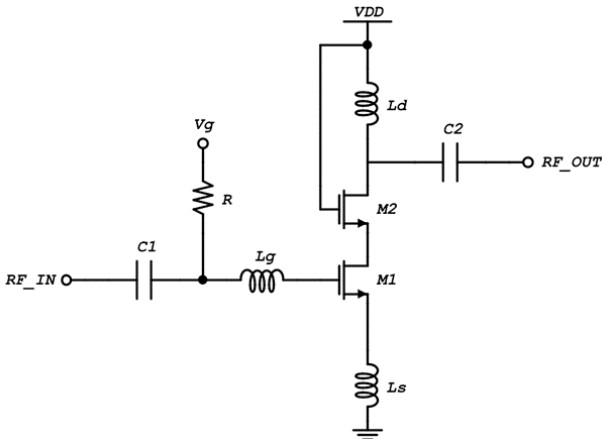

**Figure 3.** Common source LNA topology with input matching network.

To obtain the input impedance matching criteria, small-signal analysis was applied to the input part of the LNA topology. Stepping aside the DC blocking capacitor C1 and the bias circuitry, the small-signal analysis topology can be derived, as illustrated in Figure 4. The impedance of the LNA input network can be derived as

$$Z_{in} = \frac{u_{in}}{i_{in}} = j\omega L_g + \frac{u_s + u_{gs}}{i_{in}}, \quad u_{gs} = \frac{i_{in}}{j\omega C_{gs}}, \tag{4}$$

where $u_s$ is the source voltage and $u_{gs}$ is the gate-source voltage of input transistor while $i_{in}$ is the input current. The source current can be derived as

$$i_s = i_{in} + g_m u_{gs} = i_{in}\left(1 + \frac{g_m}{j\omega C_{gs}}\right), \tag{5}$$

where $g_m$ is the transconductance of the input transistor M1. According to (5), the source voltage $u_s$ can be calculated as

$$u_s = i_{in}\left(1 + \frac{g_m}{j\omega C_{gs}}\right) \times j\omega L_s. \tag{6}$$

Thus, from (4) and (6), the input impedance and the real and imaginary parts of the input impedance of the LNA topology depicted in Figure 4, can be derived as

$$Z_{in} = \frac{g_m \times L_s}{C_{gs}} + j\left[\omega(L_s + L_g) - \frac{1}{\omega C_{gs}}\right] \tag{7}$$

$$Re(Z_{in}) = \frac{g_m \times L_s}{C_{gs}} \tag{8}$$

$$Im(Z_{in}) = \omega(L_s + L_g) - \frac{1}{\omega C_{gs}}. \tag{9}$$

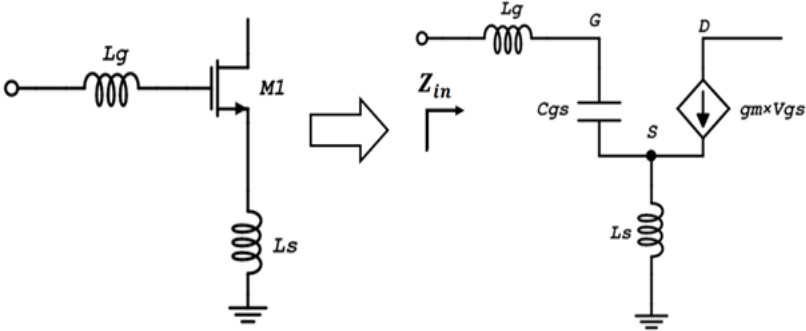

**Figure 4.** Small-signal analysis of the input network of the common source LNA topology.

The characteristic impedance of an antenna that is often connected to the input of the LNA block is 50 Ω. Thus, it is crucial, in the LNA design, to match the input impedance $Z_{in}$ of the LNA to 50 Ω to achieve full power transmission with negligible wave reflections back to the antenna. The input impedance of the LNA topology has a real part and an imaginary part, as expressed in (8) and (9), respectively. To efficiently match the input impedance of the LNA to 50 Ω, the real part of $Z_{in}$ should be equal to 50 Ω and the imaginary part of $Z_{in}$ should be equal to zero. Hence, the complex input impedance of the LNA will be transformed to a real impedance of 50 Ω, resulting in impedance matching at the input of the LNA. According to (9), the frequency in which the imaginary part of the complex input impedance is zero can be calculated as [28]

$$Im(Z_{in}) = 0 \Rightarrow \omega = \frac{1}{\sqrt{(L_s + L_g) \times C_{gs}}}. \tag{10}$$

The output impedance of the LNA is not crucial to be 50 $\Omega$ unless the next stage, that the LNA drives, needs a 50 $\Omega$ termination (an output matching network should be used in this scenario). In all other cases, the output impedance of the LNA could not be necessarily set to 50 $\Omega$. The output impedance and the gain of the LNA topology depicted in Figure 3 can be derived as

$$Z_{out} = g_{m(M2)} r_{o(M2)} r_{o(M1)} \left\| \left\| \frac{1}{j\omega C_{gs}} \right\| \right| j\omega L_d \quad , \quad A_V = g_m \times Z_{out}. \tag{11}$$

Thus, the drain inductor $L_d$ can be used to adjust the output gain of the topology to the desired value. It is not easy to adjust the output gain by varying the transconductance $g_m$ of the transistors due to the fact that the input impedance matching also depends on the transconductance $g_m$ and, by changing the $g_m$ of the topology, the input impedance will not be matched to the source impedance.

*2.3. LNA Input Matching Methodology*

To achieve a very precise input impedance matching, an LNA matching methodology using a Smith chart was applied. There are two important steps for the design of the input matching network of the LNA. The first step was to exclude the gate inductor $L_g$ from the input matching network of the LNA to obtain a 50 $\Omega$ real part of the complex input impedance at the desired operating frequency. Having (8) as reference, for a constant $C_{gs}$ (depends on the input transistor sizing) and a desired $g_m$, and by varying the source inductor $L_s$, a 50 $\Omega$ real part of the input complex impedance can be obtained, as graphically illustrated in Figure 5. It should be mentioned here that Smith charts represent normalized impedances with respect to the source impedance [29]. Hence, the normalized impedance values illustrated in the Smith chart should be multiplied by the source impedance (in this case 50 $\Omega$). The $L_s$ sweep curve intersects the 50 $\Omega$ constant resistance circle in two points (M1 and M2 points on the smith chart). The $L_s$ values at those two intersection points results in a 50 $\Omega$ real part of the input impedance. In addition, these two intersection values of the $L_s$ can be used as reference points and an $L_g$ sweep can be performed for fine tuning of the input impedance to achieve a $VSWR \leq 1.05$.

The second step of the input impedance matching network of the LNA is to zero out the imaginary part of the complex input impedance at the desired frequency. According to (10), for a constant $C_{gs}$ and by setting the value of the source inductor $L_s$ to one of the two intersection points provided in Figure 5, the imaginary part of the complex input impedance can be zeroed out by varying the $L_g$ inductor value of the LNA at the desired frequency of operation, as graphically illustrated in Figure 6. The $L_g$ value in which the imaginary part of the complex input impedance is zero, combined with the $L_s$ value in which the real part of the complex input impedance is 50 $\Omega$, derive the input matching network of the LNA at a specific operating frequency. Using this methodology, a $VSWR < 1.05$ can be achieved resulting in a perfect input impedance match and negligible wave reflections, as depicted in Figure 7.

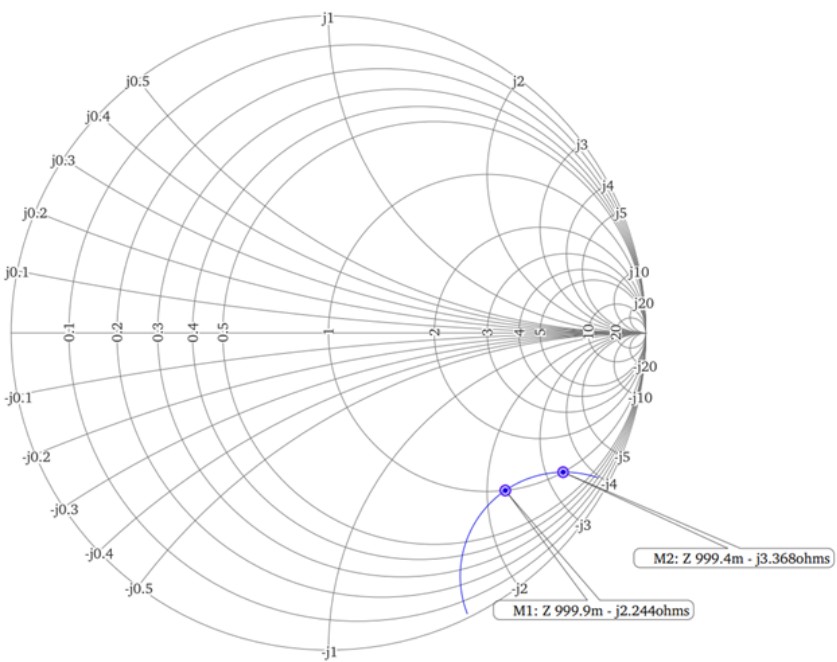

**Figure 5.** Impedance matching of the real part of the LNA's complex input impedance.

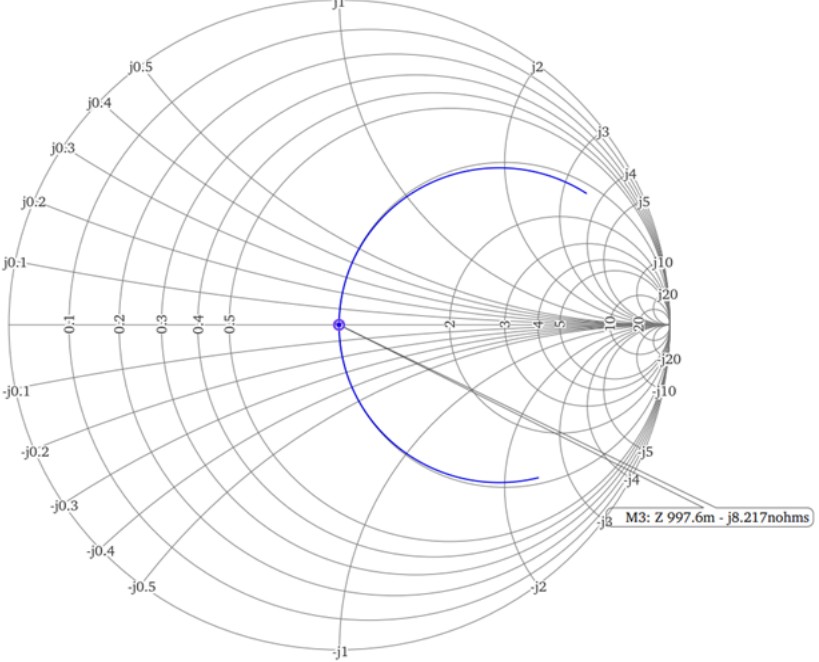

**Figure 6.** Impedance matching of the imaginary part of the LNA's complex input impedance.

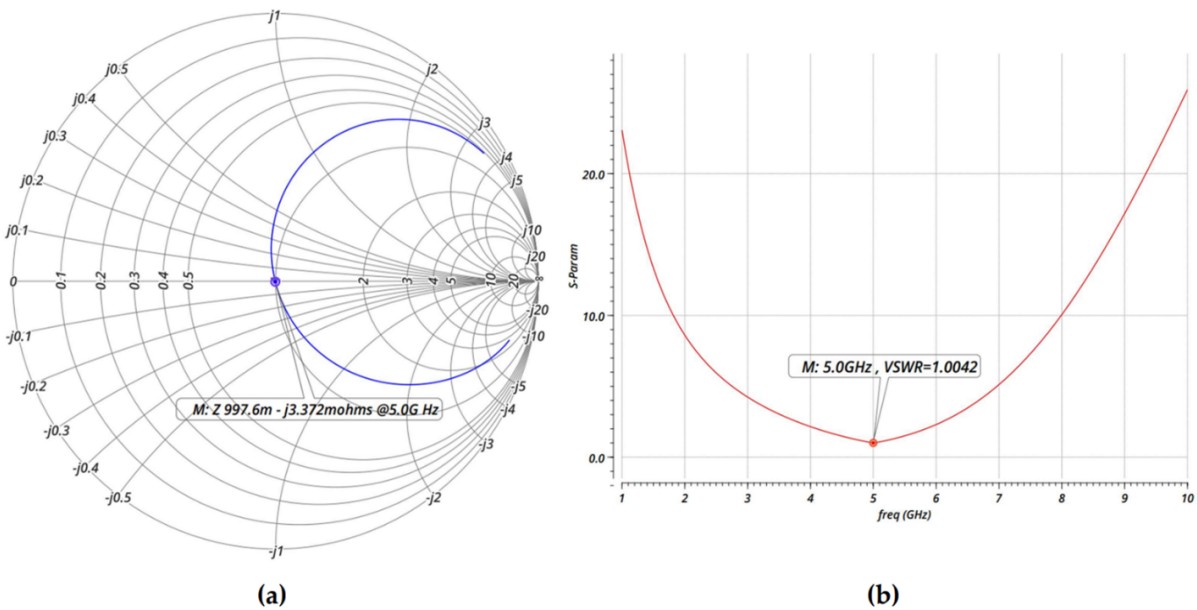

**Figure 7.** (**a**) Input impedance and (**b**) VSWR with respect to frequency, for a 5 GHz LNA design.

### 3. Hybrid Genetic Algorithm and Machine Learning Design Methodology

The proposed design tool is an end-to-end solution, which efficiently combines evolutionary search, specifically, genetic algorithm (GA) and machine learning (ML) techniques, in an automated manner to learn a Design Space and accurately predict any desired optimal design from this space. Evolutionary search is used to find optimal design parameters in an automated manner instead of the classical hand-crafted design process. Then, ML techniques are adopted to generalize the GA's results to other design cases from the under-exploration Design Space.

The following contributions to implementing the proposed automated design tool were made by the authors. As it is depicted in Figure 8, firstly, the designer describes the Design Space. Then, some points from the Design Space are sampled to build a dataset. Afterwards, for each of the sampled Design Space's points, a genetic algorithm searches the optimal design parameters. The created dataset from the above operations has the format of (Design specifications, Optimal Design Parameters). Finally, using the above data, the training dataset is created and used to train a set of predictors, taking advantage of polynomial regression. As predictors' input features, the Design Specifications were used whereas, as predictors' targets, the optimal design parameters' values were used as input specifications. The total framework includes iterations of the "sampling-GA execution-dataset creation-training-testing" cycle with the goal of a minimum-base desired accuracy for the predictors. If the desired accuracy is reached, then the iterations will finish and the whole process terminates, returning the final predictors that can be used to predict any desired optimal design. The core algorithm of the above framework is the automated sampling and dataset creation algorithm, which is described in the next sub-section and shown in Figure 9.

The parameters of the design process are classified into three categories: the technology parameters, which have fixed and predefined values according to the fabrication process of the design, the design parameters, which are used to describe the desired design specifications according to the targeted application, and last but not least, the Design Space definition and the exploration variables, which are the targets of the predictors.

The total procedure is implemented in two stages. The first stage is the pre-process, when the proposed tool learns a described Design Space. This step needs to be executed only one time for a Design Space and includes rounds of "sampling-GA execution-dataset creation-training-testing" cycle, until the desired accuracy for the predictors is reached.

Then, in the second stage, using the trained predictors from the first stage, the designer can predict any optimal design from the described Design Space.

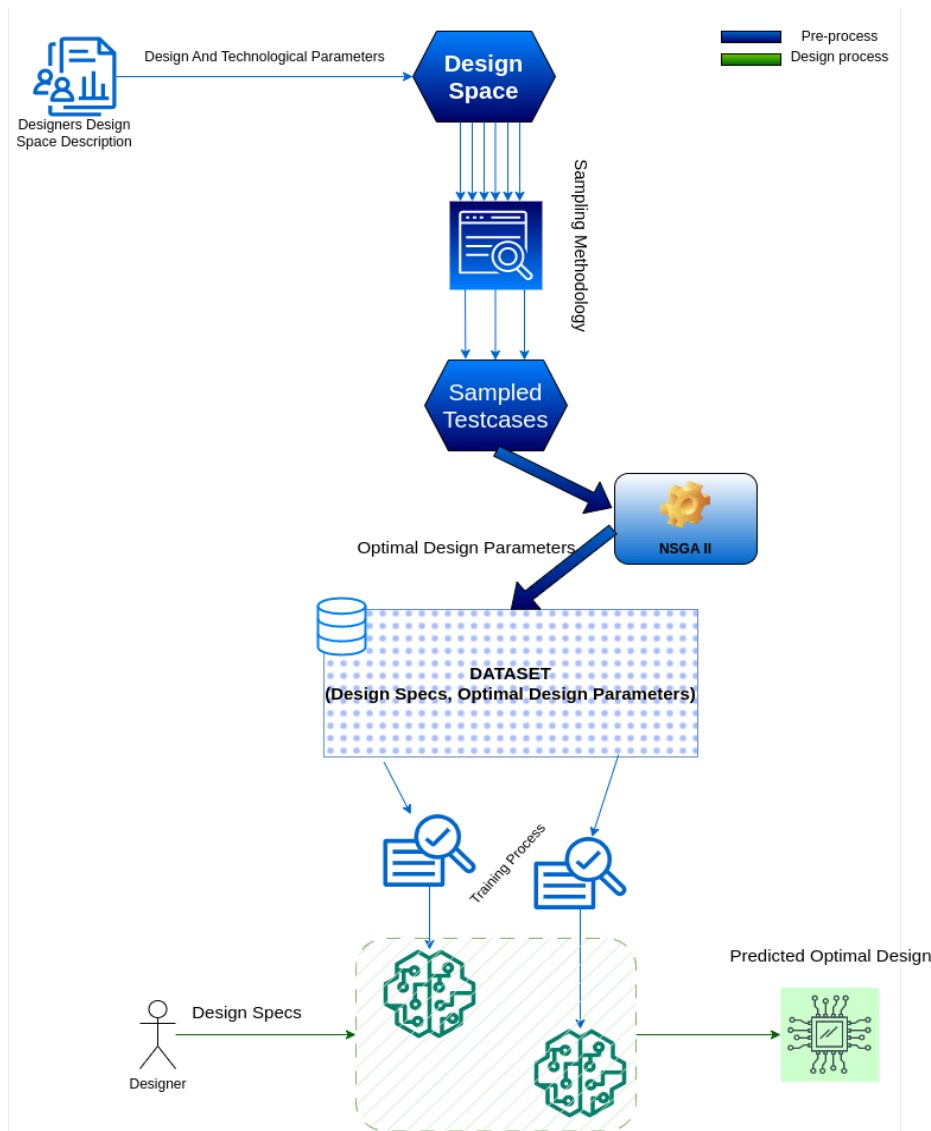

**Figure 8.** Proposed framework.

The rest of the section includes a detailed analysis of the individual parts of the hybrid tool. In particular, the implemented sampling methodology and the adopted genetic algorithm were described and, finally, the used machine learning techniques were analyzed.

### 3.1. Sampling and Dataset Creation Methodology

To build a training dataset for the predictors' training, a sampling methodology is a necessity. As an optimal number of samples, the authors considered the smallest number that ensures an accepted accuracy for the predictor, according to design specifications. The adopted sampling and dataset creation methodology is simple yet efficient and leads to very accurate predictors, using the sampled data points from the Design Space as a training set. The sampling and dataset creation operations are based on an exhaustive search. The sampler begins by sampling only two points, then the evolutionary search and the training of the predictors are implemented and the accuracy of the predictors is computed. Furthermore, the whole process is repeated including one more point on every iteration and the iterations stop when the accuracy reaches an accepted value. Analytically, the sampling and dataset creation methodology includes iterations of the "sampling-GA

execution-dataset creation-training-testing" cycle, until the desired accuracy is reached but, in each iteration, one extra point was sampled and added into the dataset.

Figure 9 describes in detail the sampling and dataset creation operation that was used. The above operation is the core operation of the proposed framework. In each case, the points were sampled from the Design Space in such a way that the Design Space was split into equal parts, as illustrated in Figure 10 for the example of a three data point split. This sample placement is preferred due to the selected polynomial-regression technique that was used for the predictor. The samples are selected to describe each Design Space in as detailed a manner as possible, considering the number of the samples, so they are placed symmetrically inside the Design Space.

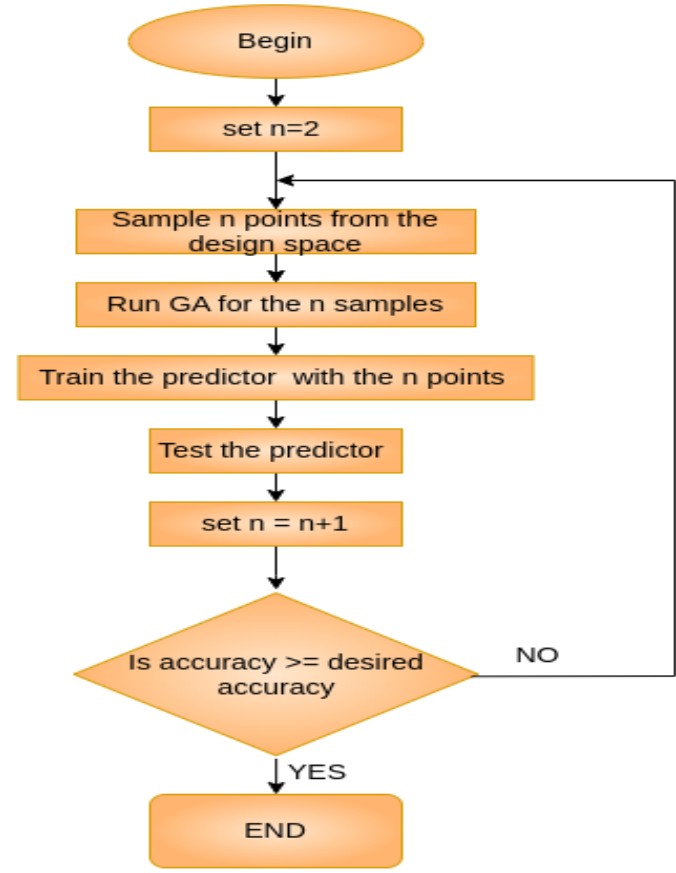

**Figure 9.** Sampling and dataset creation methodology overview.

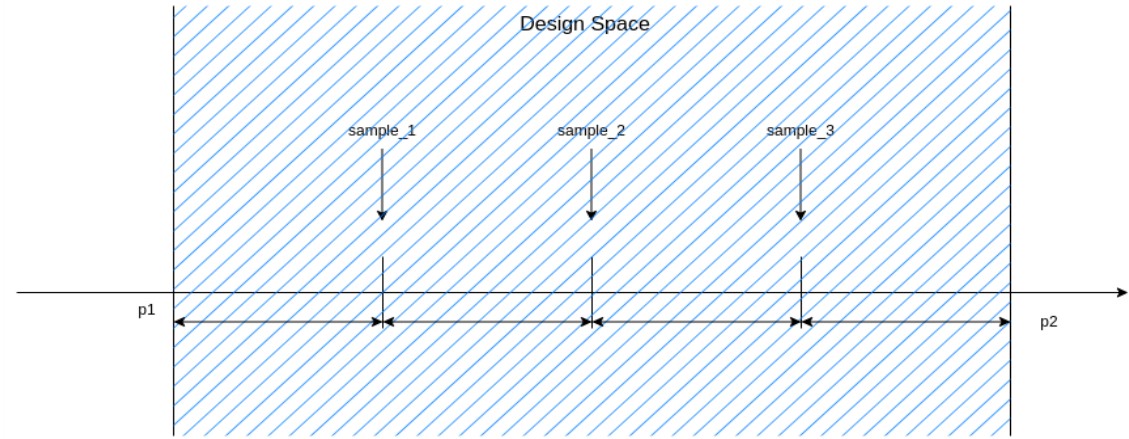

**Figure 10.** Example of sampling three points from a Design Space.

Adopting the above exhaustive search methodology, the automated dataset creation task was implemented, but also the polynomial regression parameter was defined in an automated manner. In particular, the number of dataset points were automatically selected and the polynomial grade was set to be as high as possible (one less than the number of data points).

### 3.2. Evolutionary Search and Genetic Algorithms

The problem of finding the optimal solution, in the sense of a fitness (or cost) function, is often referred to as an optimization problem. Genetic algorithms (GAs) are some of the most popular solution approaches for this category of problems. They efficiently explore the search space and converge to the optimal solution without searching all the candidate solutions. GAs are basically used to solve multi-objective optimization problems, where the target is to optimize two or more objective functions. In this category, the design process of an analog circuit can be classified, where the target is to find a combination of design parameters that optimize a set of efficiency metrics, but also taking into consideration some parameters' constraints based on the design technology that is employed. The classical multi-objective problem formulation for a double objective minimization problem with two variables is described in (12). To this end, a genetic algorithm was utilized to solve the optimization design problem for each of the sampled points from the Design Space (as it is described in the previous section).

$$
\begin{aligned}
minimize\ a_1 \times f_1(x_1, x_2) + a_2 \times f_2(x_1, x_2), s.t. : \\
xmin_1 \leq x_1 \leq xmax_1, \\
xmin_2 \leq x_2 \leq xmax_2.
\end{aligned}
\tag{12}
$$

Algorithms based on physical and natural phenomena are characterized as physical-inspired algorithms. In addition, genetic algorithms are categorized as evolutionary as they adopt the idea of searching under the impact of a fitness function with the tendency of evolving in the direction that improves it. Genetic algorithms are also classified in the category of meta-heuristics as they describe a generalized heuristic solution for a variety of problems. Being one of the most popular population-based meta-heuristics, Genetic algorithms are composed from three operations, namely selection, crossover, and mutation [30].

The total process of a GA's run is based on a population of chromosomes, where each chromosome encodes a candidate solution. Then, a sequence of iterations for the operations selection, crossover, and mutation is executed until a termination condition is reached, as illustrated in Figure 11. Specifically, new generations are created based on the adaption operation, starting from random solutions and mixing them in a random manner. A fitness function controls the whole operation, so every new generation is more likely to be better in the sense of the fitness function. One important feature of GAs is their ability to be implemented in a parallel way, significantly speeding up the whole process.

A chromosome describes a candidate solution of the optimization problem. For example, in the cases of design problems, a chromosome is implemented as a set of numbers that coincides with the number of explored parameters of the design (the parameters that their optimal values should be explored). Thence, each of the chromosome's position gets a value from all possible values of the design parameter that it encodes and, to this end, a chromosome encodes a possible solution (a possible design).

Selection is the operation of defining the couples of chromosomes that will participate in the crossover operation. In most cases, a selection tournament is implemented where a couple of chromosomes are selected randomly and, from these two chromosomes, the best one is selected as the candidate parent for the crossover operation. This process is repeated until the desired number of chromosomes, that will consist of the couples for the crossover operation, is reached.

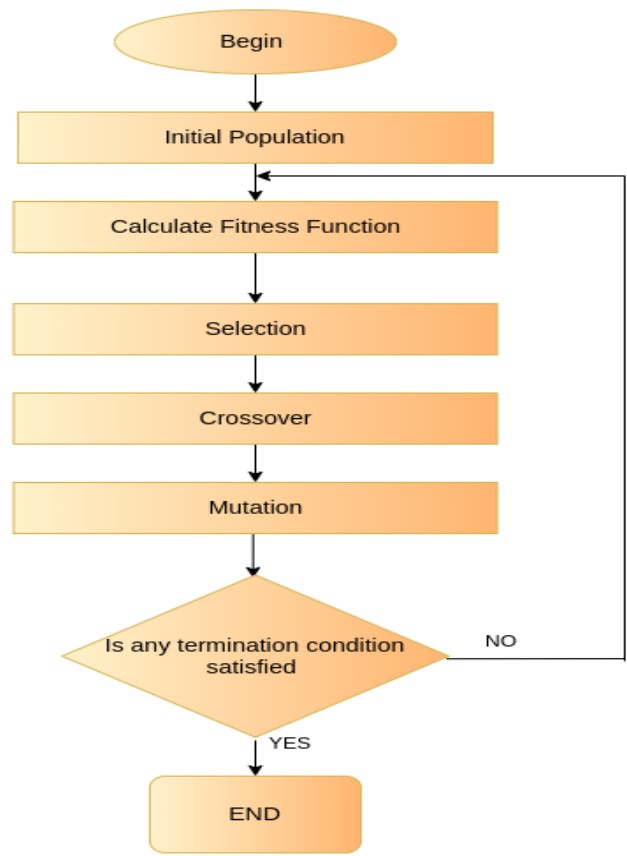

**Figure 11.** Genetic algorithm's overview [30].

Crossover is the core of a genetic algorithm as it implements the mixing of the candidate parents to create the next generation population of chromosomes. The most used crossover operation is the one-point crossover. First, a random couple from the candidate parents (from the selection tournament) is selected, a random point at the chromosomes is defined and, finally, each of the points, from the start to the random (crossover) point of the first chromosome, migrates to the second chromosome and vice versa. The mixing process of the two chromosomes (couple) is implemented with a predefined probability (in most cases >90%). In contrast, there is a small probability (<10%) for a couple to survive to the next generation without mixing and as a result without any changes. The crossover operation finishes when all candidate parents from the selection tournament are matched. Figure 12 describes the one point crossover operation for a couple of chromosomes.

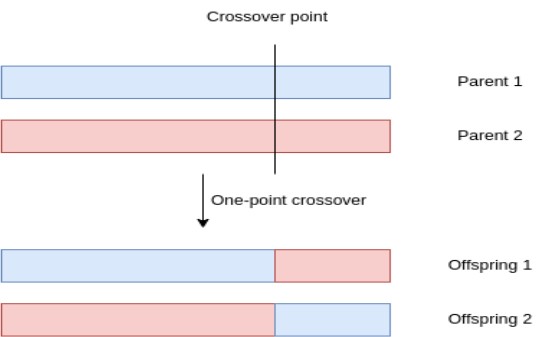

**Figure 12.** Example of one-point crossover operation [31].

Mutation is the operation that helps the genetic algorithm to avoid trapping in a local optimal solution. Each of the chromosomes has a very small probability (in most cases <2%) of changing a little. Mutation includes a part of non-ideal solutions to the population, helping the process not to trap into a local optimal solution by adding a sense of randomness in the algorithm.

A vast amount of GAs' implementations take advantage of some termination conditions, helping the meta-heuristic algorithm to decide if the optimal solution is reached or if it is trapped into a local optimal solution thus, leading to faster algorithms. Some of the well-known termination conditions are the maximum number of generations (iterations) which defines the maximum number of iterations for the GA. Additionally, in some examples, the maximum number of generations without global improvement is adopted. This termination condition describes the maximum number of iterations that the best chromosome of the current generation is worst than the global best chromosome of all the pasted generations. Finally, one more termination condition that is usually met is the maximum number of generations without local improvement. The aforementioned one refers to the maximum number of iterations in which the best chromosome of sequential generations is not improved. Probably, the most used termination condition in the literature is the first one—the maximum number of generations—but a combination of them could also be included in some GAs.

In the proposed approach, NSGA-II was selected as the multi-objective GA optimization tool. It takes advantage of a fast non-dominated sorting approach with elitism and diversity. The main contribution of the algorithm is the combination of sequential generations and then the use of a non-dominated shorting algorithm to create non-dominated fronts. Afterwards, a new population with the same number of chromosomes as at the beginning is created from the non-dominated shorted fronts. Finally, a crowding distance based shorting metric is performed to select individuals that enhance the diversity of the solutions [8].

*3.3. Machine Learning and Regression Techniques*

To generalize the results of the GA simulation-based designs for the sampled cases derived from a specific Design Space, an ML model based on polynomial-regression was employed. ML is classified in the category of Artificial Intelligence (AI) and relies on the training, validation and testing operations to learn and recognize patterns to a desired dataset. ML started attracting research interest after the perceptron architecture and back-propagation operations were proposed [32].

The problems that ML models deal with are placed into two general categories—the classification and the regression problems; hence, the classification ML models and the regression ML models both exist, respectively. The task of classification models is to classify the dataset points into several categories. In contrast, regression models try to describe one or more continuous-valued dependent variables as functions of the observations in the data [32].

To evaluate the ML approaches, some very important features were evaluated on the models. The ability to generalize well to new data is very important as well as avoiding overfitting to the training data. Overfitting describes a condition in which an ML model begins to learn the noise from the training data instead of learning just the basic tendency of the dataset and loses the ability to generalize well to new testing and validation data [32].

Additionally, ML models are categorized into supervised and unsupervised learning. Supervised learning approaches are based on the operations of training, validation, and testing and the used datasets must contain the desired solution (label). Some well-known supervised learning algorithms are linear regression, polynomial regression, logistic regression, decision trees, support vector machines (SVMs), and ANNs. In unsupervised learning, unlabeled data are used and the algorithm's main task is to group data points based on their features. Some important algorithm examples are k-means and Principal Component Analysis (PCA) and their variants [32].

The proposed ML model refers to a regression model and, specifically, a polynomial-regression model. Generally, regression methods are designed to predict continuous numeric outputs where an order relation is defined. Regression methods search for a relationship (correlation) among variables. Some well-known regression approaches are linear and generalized linear regression, polynomial regression, least and partial least squares regression (LS and PLS), least absolute shrinkage and selection operator (LASSO) and ridge regression, multivariate adaptive regression splines (MARS), and least angle regression (LARS) [33].

Linear regression is one of the most simple and most used regression techniques. Based on the assumption that the target (dependent variable) can be described linearly from the independent variables, the purpose of the training is to find the parameters of the linear function that describe the relationship between depended and independent variables. In Figure 13a, an example of linear regression with one independent variables and one dependent variable is presented. Additionally, (13) is the basic equation of linear regression model, where $y$ is the dependent variable, $x_1, \ldots x_r$ are the independent variables, $b_0, \ldots, b_r$ are the regression coefficients, and $e$ is the random error.

$$y = b_0 + b_1 \times x_1 + \ldots b_r \times x_r + e. \tag{13}$$

Polynomial regression is based on the assumption that the dependent variable is connected, in a polynomial relation manner, with the independent variables. It can be assumed to be a generalized method of linear regression and, in this case, instead of using only linear terms in the predictors equation, non-linear parts were also used. An example polynomial regression equation, with one independent variable and of second grade is presented in (14). Finally, in Figure 13b, an example of polynomial regression with one dependent and one independent variable is illustrated. It has to be mentioned that, in polynomial regression, the only non-training parameter that has to be defined is the polynomial grade.

$$y = b_0 + b_1 \times x + b_2 \times x^2 + e. \tag{14}$$

To this end, the proposed regression model used to predict the optimal variable values in our approach was a polynomial regression model, so that the non-linear relationship between the desired design specifications (independent variables) and the optimal design variables (dependent variables) could be reached from the predictor.

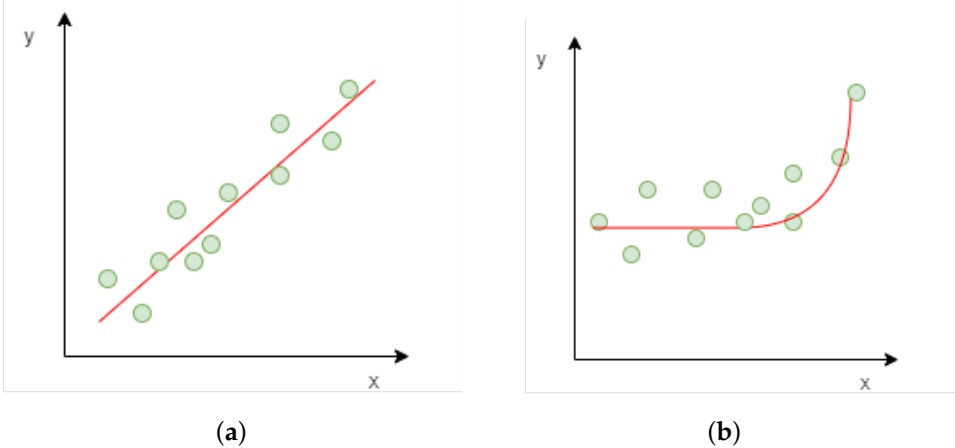

(**a**)           (**b**)

**Figure 13.** Examples of linear and polynomial regression. (**a**) Example of linear regression. (**b**) Example of polynomial regression.

## 4. Experimental Setup

The proposed framework was evaluated using a low-noise-amplifier (LNA) test case topology. In particular, the effectiveness of the methodology was evaluated, comparing its results with the baseline results. As baseline design methodology, the authors considered the classical simulation-based and human-implemented methodology (analyzed in Section 2). For evaluation purposes, two categories of metrics were considered; the first one refers to the efficiency of the designed circuit and the second to the accuracy of the trained predictors.

From the designer's perspective, using the design methodology illustrated in Figure 14, a highly-matched LNA design can be implemented but with significant time-consuming cost compared to the fully-automated ML-GA proposed methodology. Primarily, the Design Space (frequency of operation) of the LNA topology should be defined. According to the targeted frequency of the LNA, the desired topology and the optimum transistor sizing ($C_{gs}$ requirement, (8) and (9)) should be picked to be able to accurately match the input impedance of the LNA and achieve extremely low wave reflections. Furthermore, secondary design parameters, such as bias voltage of the input MOSFET and output inductor $L_d$, should be defined wisely to achieve the desired specs of the designed topology. Subsequently, $L_s$ and $L_g$ sweeps should be performed to precisely match the real and the imaginary part of the input impedance of the LNA, respectively. Finally, if the derived matching network in combination with the secondary design parameters are not adequate to meet the desired specs of the topology, a fine-tuning of the $L_s$ and $L_g$ inductors should be performed. Otherwise, the design process is finished and the designed LNA is precisely matched at the targeted operating frequency dictated by the application standards.

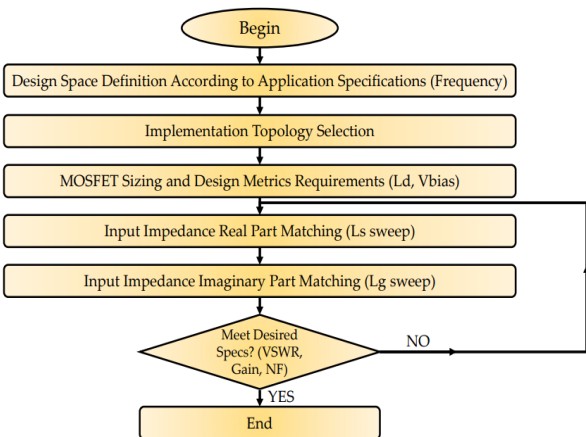

**Figure 14.** Design flow overview of a low-noise-amplifier circuit.

The Design Spaces were defined using the IEEE Radar-Frequency band designation [34] to cover a wide frequency spectrum of the wireless application portfolio. Hence, as presented in Table 1, the application spectrum was split into three major Design Spaces frequency-wise, with respect to the transistor sizing. The transistor sizing was the driving factor when the Design Spaces were divided. The optimum transistor sizing was used to cover a wide frequency spectrum and, at the same time, to reduce the design complexity by minimizing the matching-dependent parameters. The first Design Space covers all of the frequency spectrum of the *S* Band and a small portion of the *C* Band (2–5 GHz). This Design Space frequency spectrum was achieved for transistor sizing of *W* = 320 μm and *L* = 60 nm for both input and cascaded transistor of the LNA topology. The second Design Space includes three frequency bands, the *C* Band, the *X* Band and the *Ku* Band and extends from 5–16 GHz. This Design Space frequency spectrum was achieved for transistors sizing of *W* = 96 μm and *L* = 60 nm. The third and last Design Space covers the frequency range of 16–40 GHz, i.e., includes a small portion of the *Ku* Band and both *K* and *Ka* Bands. This Design Space frequency spectrum was achieved for transistors sizing

of $W = 32$ μm and $L = 60$ nm. Combining all three Design Spaces, for frequencies of 2–40 GHz, a highly-matched LNA topology can be efficiently designed using the proposed methodology. In each of the Design Spaces, a set of predictors was trained and, for any new design prediction, the suitable set of predictors is selected based on the described specifications from the designer.

**Table 1.** Design space definition and related applications.

| Design Spaces | Frequency Range | Transistors Sizing | Related Applications |
|---|---|---|---|
| Design Space 1 | 2 GHz–5 GHz | $W = 320$ μm, $L = 60$ nm | Wi-Fi, Bluetooth, WLAN Bands |
| Design Space 2 | 5 GHz–16 GHz | $W = 96$ μm, $L = 60$ nm | Radars, 5 GHz Wi-Fi Channels, Broadcasting Satellites, Aircraft Navigation |
| Design Space 3 | 16 GHz–40 GHz | $W = 32$ μm, $L = 60$ nm | Radio-astronomy, Advanced Communication Systems, Remote Sensing |

The design parameters that were used for the experiments can be categorized into two sub-sets—the spec-dependant and matching-dependant design parameters. The matching-dependant design parameters are the most crucial parameters when designing a high-frequency RF/mmWave circuit and, in the proposed methodology, the attention was driven into them. Those matching-dependant parameters are the operating frequency $f$ of the LNA, and the input matching network composed by the source and gate inductors $L_s$ and $L_g$, respectively. The frequency $f$ was used as the input parameter for the ML polynomial regression models while the $L_s$ and $L_g$ inductors were used as the exploration parameters (variables), which were the targets of the ML polynomial regression models. Furthermore, the $S_{11}$-parameters with respect to the $L_s$ and $L_g$ inductors, were used as a cost function for the genetic algorithm (GA), targeting to minimize them for better input impedance matching (low wave reflection coefficient target).

The spec-dependant design parameters, such as the drain inductor $L_d$ and the input bias voltage $V_g$, are responsible for tuning the secondary features of the topology by adjusting the gain, the noise figure, the linearity, etc., of the LNA design to meet the desired specifications. However, those spec-dependant parameters have a grave impact on the input impedance matching of the LNA as well. Due to this design complexity, the spec-dependant parameters $L_d$ and $V_g$ were static across all Design Spaces that were used in the design process. Their optimization and impact on the final LNA design was not targeted by the proposed ML-GA design methodology. The $L_d$ and $V_g$ values, in combination with the sizing of the transistors, were set so as to achieve a reasonably satisfying LNA performance across all three Design Spaces, regarding the gain, the noise figure, and the linearity of the design. Hence, the static parameters, $L_d$, $V_g$, and $V_{DD}$, the dynamic parameters with respect to the application frequency $f$, $L_g$ and $L_s$ and the design specifications across all three Design Spaces, were presented in Table 2.

**Table 2.** LNA specs across all three Design Spaces.

| Matching-Dependant Parameters | Spec-Dependant Parameters | Parameter Values | LNA Specs Across Design Spaces |
|---|---|---|---|
| $f$ | $L_d$ | 500 pH | $Gain > 12$ dB |
| $L_g$ | $V_g$ | 500 mV | $NF < 2$ dB |
| $L_s$ | $V_{DD}$ | 1 V | $IP_{1dB} > -10$ dBm |

From the methodology perspective, in the preprocess stage, the design parameter values and the desired frequency range, so as to completely describe the under-exploration Design Space, were defined by the designer. Then, the sampling process was implemented where a number of frequencies were sampled (as described in Section 3.1) from the desired range. For each of the sampled frequencies ($f_i$), a GA run was implemented to find the optimal ($L_s, L_g$) values that minimize the efficiency parameters ($S_{11}$), considering that $f = f_i$ and that the design parameters of the circuit have the fixed predefined values. The cost function for the GA iterations was the $S_{11}$ parameter and it was computed using SPECTRE-based simulations. Additionally, the GA adopted for this part of the framework is NSGA-II and all the parameters were set at their default values based on the NSGA-II documentation. The only GA parameters that were tuned are, the number of the generation and the number of chromosomes per generation (population size) which were set at 200 and 20, respectively.

After finishing the GA runs for all the sampled frequencies, the training dataset was completed. The training dataset was in the form of ($f, (L_s, L_g)$), where $f$ is the frequency and $L_s$, $L_g$ are the optimal inductor values for the specified frequency. To finish the training process, two polynomial regression models were trained with this dataset. The only parameter for the polynomial regression models was the polynomial grade, which was defined from the number of the training points. The grade was the highest possible based on the number of training points i.e., $grade = \#training\,points - 1$. Both models used frequency as an input feature and the first model predicted the optimal $L_s$ as soon as the second one predicted the optimal $L_g$ value.

Finally, for testing purposes, some intermediate frequencies were sampled and GA runs were implemented to find the optimal $L_s$, $L_g$, respectively, and a testing dataset was created and used to compute the accuracy of the predictors. The testing set for each of the described Design Spaces was created only once at the beginning of the framework execution. The test sets sizes for Design Space 1, Design Space 2, and Design Space 3 were 30 points, 110 points, and 50 points, respectively. The test sets' sizes were exclusively derived from the design specifications of the LNA. At high operating frequencies (Design Spaces 2 and 3), there is no need to design an LNA using two decimal points in the frequency specification aspect because this kind of accuracy is impossible to achieve after the physical design fabrication in silicon.

This whole process was repeated until the termination condition was verified (the desired accuracy was reached). After the termination of the total execution, a set of predictors was successfully trained to predict the optimal ($L_s, L_g$) values, for any frequency of the desired Design Space. Analytically, the proposed framework adopts three termination conditions. The first one is the local minimum check, which tests if the proposed predictors (with the selected number of samples) have the minimum (locally) errors, between the previous and next predictors (with one more and one less dataset points). Additionally, the second one is about the absolute value of the error metric. The adopted error metric on the framework is the Normalized Mean Square Error (NRMSE) metric, described in the (15) and (16). Hence, the termination condition was set to reach an $NRMSE < 0.1$ for both the $L_g$ and $L_s$ predictors. Finally, the last termination condition was the maximum number of total iterations, which was set at 20. The last condition was not reached in any of the tested Design Spaces.

$$RMSE = \sqrt{\sum_{i=1}^{n} \frac{(y_i - \hat{y}_i)^2}{n}} \tag{15}$$

$$NRMSE = \frac{RMSE}{y_{max} - y_{min}}. \tag{16}$$

## 5. Experimental Results

The proposed methodology was evaluated using two perspectives, the efficiency of the predicted circuits and the accuracy of the predictors. For the first perspective, after the whole sampling-optimization-training process was finished, some characteristic use-cases (frequencies) were selected and their optimal design was produced using the classical methodology (SPECTRE-based simulations) by the designer. Then, the base-line circuit was compared with the predicted one from the proposed predictors in the sense of efficiency metrics. Due to the fact that the proposed design framework presented in this work attempts to create highly matched LNA circuits, the efficiency metric used in the ML-GA framework comparison was the voltage-standing-wave-ratio or VSWR. The VSWR monitoring is a necessity when designing LNAs or any kind of RF/mmWave circuitry to accurately estimate the wave reflections due to mismatches between input and load impedances. In these cases where the impedance matching is the efficiency metric, it is unfair to make a straight-forward comparison of the $S_{11}$ parameters derived by simulating every designed test case. In some parameter sets of $(f, L_g, L_s)$, the $S_{11}$ parameter derived from the ML-GA methodology can be equal to $S_{11} = -35$ dB, whereas the $S_{11}$ parameter derived from SPECTRE-based simulations can be equal to $S_{11} = -70$ dB. Although these $S_{11}$ parameters have 100% deviation from each other, the $VSWR < 1.05$ in both cases, which indicates a highly matched input impedance network with negligible wave reflections, both in the ML-GA and the SPECTRE-based designs. For the second perspective, after the training process, the accuracy of each predictor was computed on the composed test set, having as reference the $f$, $L_g$ and $L_s$ parameter values derived from SPECTRE-based simulations.

The simulation results derived from the proposed ML-GA analog design automation framework and from SPECTRE-based RF simulations are depicted in Figure 15. In each Design Space, for frequencies in the range of 2 GHz–40 GHz, the $L_s$ and $L_g$ accuracy comparison is illustrated while the efficiency of each predicted/generated design is monitored using the VSWR and the reflection coefficient Γ metrics.

The $L_s$ and $L_g$ predictors' Mean Absolute Error (MAE), for Design Space 1 ($W = 320$ μm and $L = 60$ nm) was calculated as $MAE = 2.55\%$ for the $L_g$ predictor and $MAE = 1.67\%$ for the $L_s$ predictor. For Design Space 2 ($W = 96$ μm and $L = 60$ nm), the mean absolute error was calculated as $MAE = 0.31\%$ for the $L_g$ predictor and $MAE = 0.63\%$ for the $L_s$ predictor. Finally, for Design Space 3 ($W = 32$ μm and $L = 60$ nm), the MAE was calculated as $MAE = 0.13\%$ for the $L_g$ predictor and $MAE = 0.39\%$ for the $L_s$ predictor. From the accuracy perspective, an overall prediction error $< 1\%$ of the $L_g$ and $L_s$ inductor values across all three Design Spaces was achieved using the proposed ML-GA design methodology.

The most fair, straight-forward, and accurate way to compare the matching efficiency of each LNA design generated/predicted using the proposed ML-GA design methodology is by monitoring the VSWR. The VSWR measurements should not be compared against the measurements derived from SPECTRE-based simulations but a VSWR limit should be defined according to the targeted reflection coefficient (efficiency). Due to the highly matched LNA designs that were targeted by the proposed methodology, the VSWR limit was set to $VSWR \leq 1.05$ to achieve >99.9% impedance match efficiency and <0.1% wave reflections. As illustrated in Figure 15, all of the predicted/generated LNA designs using the proposed ML-GA methodology have a matching efficiency > 99.9% by achieving a $VSWR < 1.05$ across all three Design Spaces. The VSWR of each Design Space achieves a local minimum (best efficiency) at every GA sampling point. Between the GA sampling points, the VSWR measurements were derived from the ML predictor $(L_s, L_g)$ sets. For both approaches, GA and ML, the VSWR specifications for highly matched LNA designs were met.

Additionally, the efficiency of the proposed sampling and dataset creation methodology was tested. As described in Section 3, the proposed sampling and dataset creation methodology is based on an exhaustive search, beginning with two sampled points for the

training dataset and adding one extra point at each iteration targeting an ideal accuracy. To evaluate the effectiveness of the proposed methodology, some extra experiments were implemented after the framework basic experimentation.

More specifically, in Figure 16, the impacts of the training dataset size and the polynomial grade are depicted. The Normalized Root Mean Square Error (NRMSE) ((15) and (16)) is used as the error metric. To compute the impact of the number of the training samples, at the predictors' accuracy, the maximum feasible polynomial grade (one less than the number of the samples) was considered as the grade of the predictors. As shown in Figure 16, adding points at the training set originally reduces the testing error but, after some points, the predictor loses the ability to be generalized and the testing error increases. For example, in the Design Space 1, the proposed framework automatically selected the four sampling points for the training set which, as illustrated in Figure 16, is the optimal one. Furthermore, in Design Space 2, the selected number of samples was nine, although a better choice of six or seven points could be selected, but the framework was not able to terminate due to strict termination conditions. Design Space 2 was more complex due to the fact that the two predictors ($L_g$, $L_s$) had different optimal number of samples but again, the framework was able to reach a paret-optimal selection (nine sampling points). Finally, in Design Space 3, the framework selected six sampling points which actually is the optimal choice for both of the predictors.

Finally, the proposed methodology initialized the polynomial grade of the predictor in an automated manner. In particular, the polynomial grade was selected to be one less than the number of sampled training data, as in most cases that is the maximum grade that the dataset is able to be trained on correctly due to the number of training parameters which increases as the polynomial grade increases. The above is also clearly in agreement with the results in Figure 16. In detail, the number of training data was set at a fixed value in any case, as described in Table 3, and then predictors with different polynomial grades were trained and tested with the same data. As illustrated in the Figure 16, in the first Design Space, the best choice for the polynomial grade was one less than the number of samples (Poly grade 3 for Design Space 1), as the NRMSE decreased at its minimum value with this predictor. For Design Spaces 2 and 3, the optimal polynomial grade was not actually the same as the framework's choice (one less than the number of samples) but the difference did not affect the predicted designs' efficiency in practice. Analytically, for Design Space 2, the polynomial grade 6 was slightly better than the selected polynomial grade 8 ($\approx$16% difference on the NRMSE). Similar, for Design Space 3 the polynomial grade 4 was slightly better than the framework's choice of poly-grade 5 ($\approx$11% difference on the NRMSE) but this did not practically affect the efficiency of the predicted design. Table 3 shows the final choices of the framework of the number of samples and the polynomial grade for all the three Design Spaces.

**Table 3.** ML parameters.

| Design Space | Number of Training Data | Polynomial Grade |
|---|---|---|
| Design Space 1 | 4 | 3 |
| Design Space 2 | 9 | 8 |
| Design Space 3 | 6 | 5 |

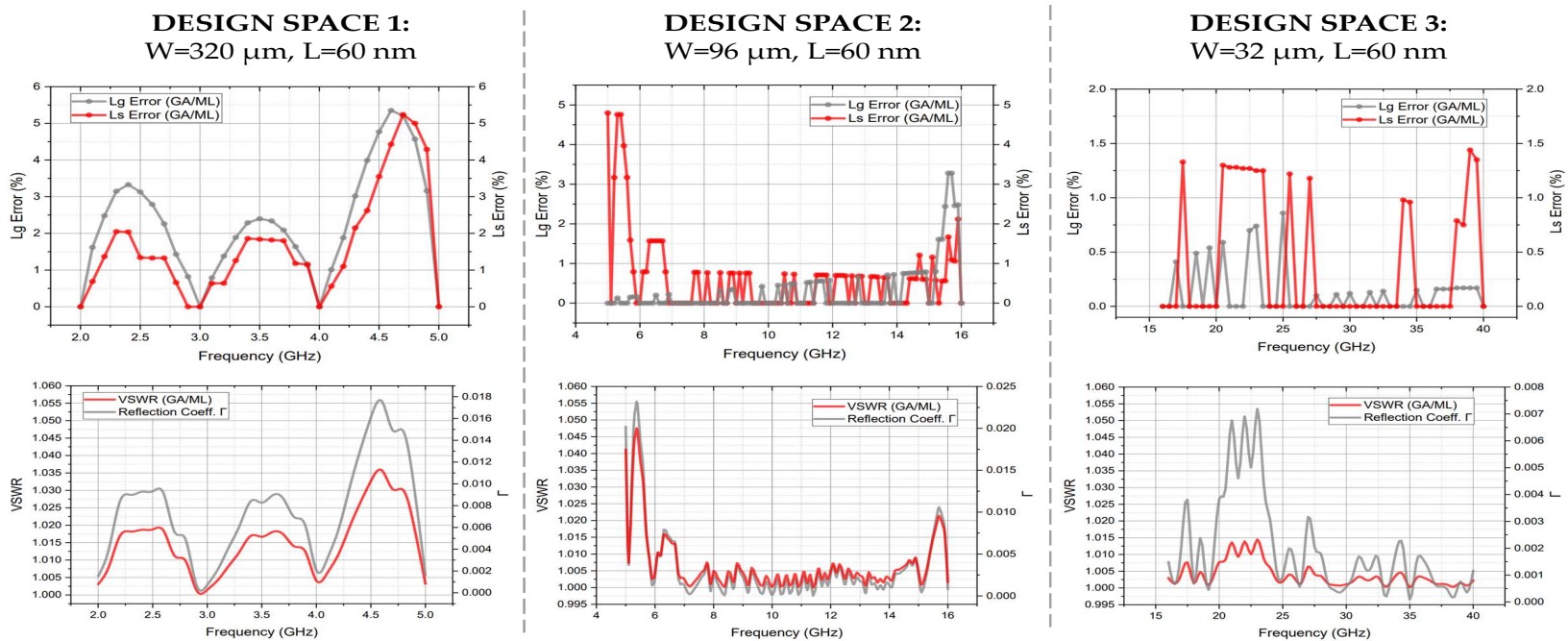

**Figure 15.** ML-GA proposed framework versus SPECTRE-based simulations: $L_s$ and $L_g$ accuracy comparison and VSWR efficiency monitoring across all three Design Spaces.

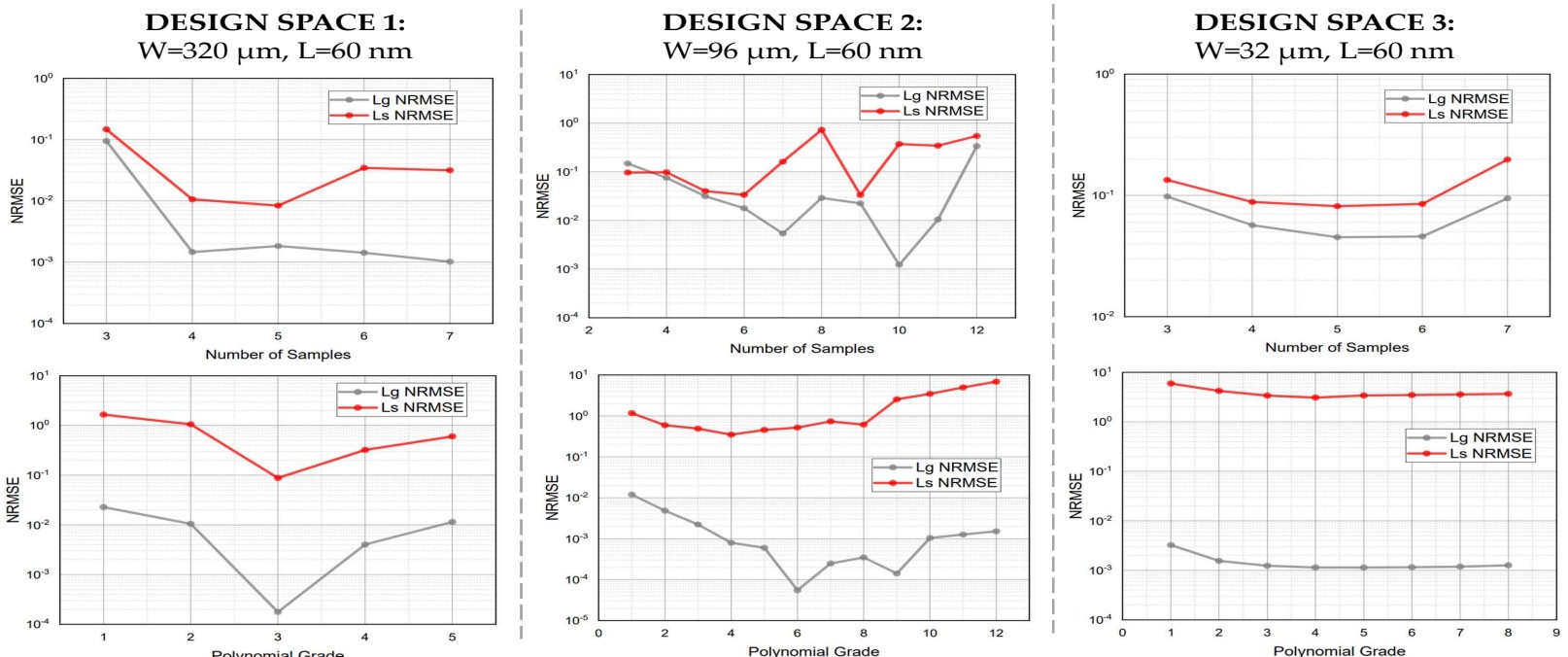

**Figure 16.** Impact of the number of samples and the polynomial grade at the predictors accuracy: *NRMSE* of $L_s$ and $L_g$ for different number of samples and for different polynomial grades.

In the last part of this section, an execution time analysis is provided to underline the superiority of the proposed hybrid approach compared to the base-line (SPECTRE-based simulations) design methodology. Table 4 provides the execution time per Design Space for each part of the proposed algorithm. More specifically, the most time consuming part is the pre-process step, which contains the dataset creation part but it is only be executed one time per Design Space. Then, the training part and the prediction part have almost no time cost, as the execution time is $\ll 1$ s. In total, the framework execution time for all the three Design Spaces and for the design prediction of all the test cases was approximately less than 20 h. On the other hand, the same process for all the Design Spaces and all the test cases, with the base-line (SPECTRE-based simulations) design methodology took approximately one week to be implemented, proving that the proposed framework tackles both the automation and acceleration tasks successfully.

**Table 4.** Execution Times.

| Design Space | Dataset Creation Time | Training Time $L_s$, $L_g$ | Prediction Time |
|---|---|---|---|
| Design Space 1 | 4 h | 0.000299, 0.000107 s | 0.00068 s |
| Design Space 2 | 9 h | 0.0003676, 0.0001476 s | 0.004941 s |
| Design Space 3 | 6 h | 0.0003089, 0.000144 s | 0.001499 s |

## 6. Discussion

Under this final section, some aspects of the proposed design methodology are discussed. The proposed methodology is end-to-end automated and dominates the GA-based approximate methodologies and the ML-based prediction methodologies. In Table 5, a brief comparison of the proposed algorithm with some related methodologies and algorithms, which are analyzed in the Introduction, is described. The first four frameworks are GA-based and, thereby, use a GA to solve the optimization problem that describes the optimal design purpose. This algorithms automate the design process, as the designer only describes the design specifications and the optimization tool (GA) approximately solves the optimization problem, co-working with a simulation tool, leading to a near optimal or near pareto-optimal design. The above algorithms refer to a single design case and thus, for a new design with different specifications, a re-execution of the GA must be implemented. The GA does not generalize the results of a single design to other designs and, thus, the speed up for a big set of design specifications is low (compared with our test cases with $\geq 40$ designs per Design Space). The generalization ability, in the comparison provided in Table 5, refers to the ability of the frameworks to generalize the results produced for a single design or some specific designs with desired specifications, to other designs with different specifications.

On the other hand, the next three approaches are ML-based, so they are able to generalize results to new design specifications. However, in this case, a dataset for training and a dataset for testing purposes need to be implemented. In all the above methodologies, the datasets were created by a designer with the base-line flow (simulation-based), so these methodologies are not fully automated. Additionally, the dataset creation is time-consuming and thus, as more data points contribute to the dataset, the total implementation time increases, resulting in a low-to-medium speedup for a big dataset of design specifications. It was clearly stated that, the prediction time is very small, in most cases less than a few seconds and also, the dataset creation process and the training process are both implemented only one time per Design Space. In addition, in the last two cases, the ML predictors are used to predict efficiency metrics for a desired design and hence, to complete the design, a designer is needed to use the frameworks instead of the simulation tool. It is obvious that the above frameworks lack of automation due to the need of designer's supervision in the whole process. In the example of [13], instead of the designer, a GA is also adopted to select the optimal design parameters, using the ML predictor results as a

cost function for the GA. In the same manner, in this particular case, the GA run needs to be executed for any different design, resulting in a low acceleration disadvantage.

**Table 5.** Methodologies comparison.

| Methodology | Test Case | Speed Up | Dataset Creation (for ML) | ML Parameters Tune (for ML) | Generalization |
|---|---|---|---|---|---|
| Baseline | Baseband/RF/mmWave | NO | - | - | ✗ |
| GA [7] | RF-LNA | LOW | - | - | ✗ |
| NSGA-II [8] | RF-LNA | LOW | - | - | ✗ |
| GA [9] | RF-LNA | LOW | - | - | ✗ |
| GA-SA-LM [10] | RF-LNA | LOW | - | - | ✗ |
| ANN-GA [11] | RF-LNA | MEDIUM | Hand-made | Automated (GA) | ✓ |
| Bayesian Linear Regression and SVMs [12] | Baseband/RF Amplifiers | MEDIUM | Hand-made | Hand-made | ✓ |
| ML-GA [13] | Baseband Op. Amps. | MEDIUM | Hand-made | Hand-made | ✓ |
| **Ours GA/ML** | **RF/mmWave LNA** | **HIGH** | **Automated (GA)** | **Automated (GA)** | ✓ |

To this end, our proposed methodology takes advantage of the automation ability of the GA to create, in a faster and automated manner, a training and a testing dataset. After the dataset is created, the proposed framework exploits an ML technique (polynomial regression) to generalize the GA results for the training dataset to new design specifications from the Design Space. In our methodology, both the dataset creation and ML parameters tuning were performed automatically, implemented with the use of the GA. The above processes were implemented only one time per Design Space. Finally, the design process was a simple prediction from the trained predictors, resulting in a high acceleration advantage for a big dataset of design specifications compared to the baseline (simulation-base) design flow. The proposed methodology fully automates and successfully accelerates the whole dataset creation-training-prediction process, enabling the design cycle speed-up for the analog and RF/mmWave design flows.

Finally, an extension of the proposed methodology for more complex and nonlinear circuits can be derived. In such scenarios, the efficiency of the predicted design is highly affected by the predicted design parameters. The accuracy of the predictors is very important for nonlinear circuits designs due to their complexity (unstable DC operating point or non-periodic behavior). The proposed polynomial grade selection of this framework (one less than the number of samples) includes the danger of over-fitting, thus enhancing the learning noise from the dataset points. In the adopted test-cases, this danger does not practically exist due to small number of samples (<10) and small polynomial grades in all explored Design Spaces. Additionally, in the LNA design test case, the main task of highly matched designs (VSWR < 1.05 ) was easily reached with the adopted methodology. To this end, an upgrade of the proposed framework for more complex and nonlinear designs can be provided in the future. More specifically, an extra exploration routine regarding the optimal polynomial grade of the predictors, in combination with the ability for the predictors to use different and independent polynomial grades adapted to each predictor, could increase the predictors' accuracy and create more efficient designs.

## 7. Conclusions

A novel RF/mmWave analog design methodology flow through developing hybrid machine learning—a genetic algorithm fully-automated design framework—was presented in this work. The presented design methodology achieved high-accuracy parameter predictions and high-efficiency design generations in a very time-conservative manner, resulting in significant design cycle acceleration.

The provided design methodology contained two stages, starting with the preprocess stage where GA optimization was adopted to find sets of optimal design parameters

and build a training set in the form of (design specifications, optimal design parameters). Then, a combination of ML polynomial regression models were trained on this dataset for the prediction of the optimal design parameters, using as input features the design specifications. The proposed framework was implemented and tested on the task of a low-noise-amplifier circuit using a 65 nm CMOS process node. Highly matched and efficient LNA circuits, in a wide frequency spectrum, were successfully generated by the proposed design methodology, achieving >99.9% wave transmission efficiency and >99% accuracy at the task of the optimal $L_s$ and $L_g$ prediction.

The framework provided in this work automates the dataset creation process in combination with the ML parameters' selection process with the use of the GA, whereas it automates and accelerates the design process with the use of ML predictors for the optimal design parameters predictions. The proposed methodology successfully combines the GA and ML approaches and automates both the dataset creation process and the design process, enabling the acceleration of the analog design flow. A further extension of the proposed methodology is feasible for more sophisticated, non-linear, or periodic state designs, using a more complex sampling methodology.

**Author Contributions:** Conceptualization, T.N. and C.S.; methodology, C.S. and A.M.; software, C.S.; validation, C.S. and A.M.; data curation, C.S. and A.M.; writing—original draft preparation, C.S. and A.M.; writing—review and editing, T.N. and K.S.; supervision, T.N. and K.S.; All authors have read and agreed to the published version of the manuscript.

**Funding:** This research has been co-financed by the European Regional Development Fund of the European Union and Greek national funds through the Operational Program Competitiveness, Entrepreneurship and Innovation, under the call RESEARCH—CREATE—INNOVATE (project code: T2EDK-01681).

**Conflicts of Interest:** The authors declare no conflict of interest.

## Abbreviations

The following abbreviations are used in this manuscript:

| | |
|---|---|
| ADA | Analog Design Automation |
| GA | Genetic Algorithm |
| ML | Machine Learning |
| NSGA-II | Non-Dominated Sorting Genetic Algorithm II |
| NN | Neural Networks |
| ANN | Artificial Neural Networks |
| LM | Levenberg-Marquardt |
| SA | Simulated Annealing |
| LNA | Low Noise Amplifier |
| RF-LNA | Radio-Frequency, Low Noise Amplifier |
| VSWR | Voltage Standing Wave Ratio |
| AI | Artificial Intelligence |
| LS | Least Squares Regression |
| PLS | Partial Least Squares Regression |
| LASSO | Least Absolute Shrinkage and Selection Operator |
| MARS | Multivariate Adaptive Regression Splines |
| LARS | Least Angle Regression |
| NRMSE | Normalized Root Mean Square Error |
| MAE | Mean Absolute Error |

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
