# Peer review of "A Hybrid GA/ML-Based End-to-End Automated Methodology for Design Acceleration of Wireless Communications CMOS LNAs"

_electronics, doi:10.3390/electronics12112428_

Round 1

Reviewer 1 Report

The work is very interesting, and several details should be introduced.

1) The framework of hybrid GA/ML method should be given, and how is applied to the LDA design.

2) The quantitative description for the proposed method should be introduced in Abstract Section.

3) Some figures are too large.

4) Details for the framework in Figure 8 should be highlighted.

1) Abstract should be revised with  quantitative analysis.

2) it is wrong for the Abbreviations: mean absolute error (MEA) in Line 552.

3) The English spelling and grammer should be checked through the manuscript.

Reviewer 2 Report

In this paper, a new methodology for the RF/mmWave analog design process, automation, and acceleration is presented. The proposed framework aims to accelerate the design cycle of analog/RF circuits by creating a dataset with Genetic Algorithm (GA) technique in a fully automated manner and training a combination of Machine Learning (ML) models for the optimal design parameters' prediction. The proposed framework has been tested on the task of a Low-Noise-Amplifier circuit design using a 65 nm CMOS process node, and the results show that it can significantly speed-up the design cycle while maintaining high performance.

The authors should make changes to address the following comments:

1.     Line 509-511, Page 18: More details about the testing dataset are desired, at least the number of testing points.

2.     Line 603-606, Page 20: The statement that “poly grade 8 for the Design Space 2 is the best choice for the polynomial grade” is confusing. More specifically, for example, smaller NRMSE for both Ls and Lg can be obtained when the poly-grade 6 is chosen for Design Space 2 as shown in Figure 16.

3.     The polynomial grade of the predictor is chosen in an intuitively inappropriate way, because high-order interpolation is generally not recommended. First of all, as illustrated in the Figure 16, “one less than the number of samples” is usually not the best choice. On the other hand, “one less than the number of samples” converts polynomial regression into polynomial interpolation, which will learn all noise from the training data, and may result in overfitting, as mentioned in Line 379, Page 14. As a bit of advice, the authors can even test all the polynomial grades and choose the best one, since the training part and the prediction part have almost no time cost, as shown in Table 4.

4.     Table 3, Page 22: It’s not necessary to keep the polynomial grades of the two polynomial regression models consistent.

5.     There are typos and grammatical errors throughout the manuscript, for instance, the abbreviation of mean absolute error in Line 552, Page 20.

Moderate editing of English language is required.

Reviewer 3 Report

It is a nice work. It is suggested to add another table to the manuscript to compared the performance of the designed LNA with previous works or some commercial products, not just the design methodology (or add some columns to table 5).

Kindly provide a more detailed explanation of the modifications needed for the proposed method to be generalized. The claim that this method can be applied to complex and nonlinear designs should be supported by demonstrating its effectiveness in another design realization.

There are some English mistakes such as "...in an low acceleration disadvantage" in line 654. It is suggested to go through the manuscript more carefully.

Round 2

Reviewer 2 Report

The paper is recommended to be accepted.